# Cutting Edges: Professional Hierarchy vs. Creative Identity in Nicolas de Launay's Fine Art Prints

**Tamara Abramovitch**

Art History Department, Faculty of Humanities, Hebrew University of Jerusalem, Jerusalem 9190501, Israel; tamara.abramovitch@gmail.com

**Abstract:** In 1783, Nicolas De Launay copied Les Baignets by Jean-Honoré Fragonard, stating it was made "by his very humble and very obedient servant", an evidence of the hierarchical tensions between painters and printmakers during the eighteenth-century. However, De Launay's loyalty is not absolute, since a critical artistic statement is found at the edge: an illusory oval frame heavily adorned with leaves and fruits of Squash, Hazelnuts, and Oak. This paper wishes to acknowledge this meticulously engraved frame, and many more added to copies throughout De Launay's successful career, as highly relevant in examining his 'obedience' and 'humbleness'. With regard to eighteenth-century writings on botany and authenticity, and to current studies on the print market, I offer a new perspective in which engravers are appreciated as active commercial artists establishing an individual signature style. In their conceptual and physical marginality these decorations allow creative freedom which challenges concepts of art appropriation and reproduction, highly relevant then and today.

**Keywords:** Nicolas de Launay; engraved frames; eighteenth-century printmakers; trade cards; professional hierarchy; art and botany; art and economic politics; Self-Marketing

## 1. Introduction: Très Humble et très Obéissant Serviteur

In 1783, the Paris engraver and publisher Nicolas de Launay (1739–1792) published an engraving after *Les Baignets*[1] by the celebrated painter Jean Honoré Fragonard (Figure 1). The phrase "by his very humble and very obedient servant", added by de Launay to the engraving, suggests the hierarchical tension between painters and printmakers during the long eighteenth century.[2] However, a comparison between the print and the original drawing (Figure 2) reveals that his dutiful pledge is not absolute. The engraver introduces a few notable changes to Fragonard's scene: he redefines the interior space, alters the mother's dress to reveal her breasts, and adds an illusionistic tree that grows out of the fireplace. Moreover, he adds his own unique calling card in the form of an oval frame decorated with oak branches, hazelnuts and squash.[3] This framing device was one of many unique frames de Launay added to prints he published after works by other artists (Lefrançois 1981, pp. 137–38).

De Launay, who was one of Fragonard's formal engravers and his friend (Rosenberg 1987, p. 418), credited the painter as the inventor and delineator of the original work, making it certain he copied the drawing.[4] Fragonard used to draw non-conventionally to create independent pieces, and made most of his drawings not before but after completing the painting, mainly for commercial use (Dupuy-Vachey 2016, p. 16). With the alterations, de Launay transforms Fragonard's original drawing into a new work, but one that he cannot formally claim fully as his own. It is important to acknowledge the dichotomy between de Launay's deferential declaration in relation to Fragonard, and his own original contributions. This article looks at de Launay's seemingly peripheral additions, which have been largely ignored in the research, and asks what role the frame plays in relation to the central image and in the overall work. It considers the engraver's changes and additions vis

à vis the work's reception and aesthetics. Most importantly, it looks at these additions as an important source for learning about the developing print market and the creative freedom and role of the engraver/publisher as a reproductive artist in the eighteenth century.

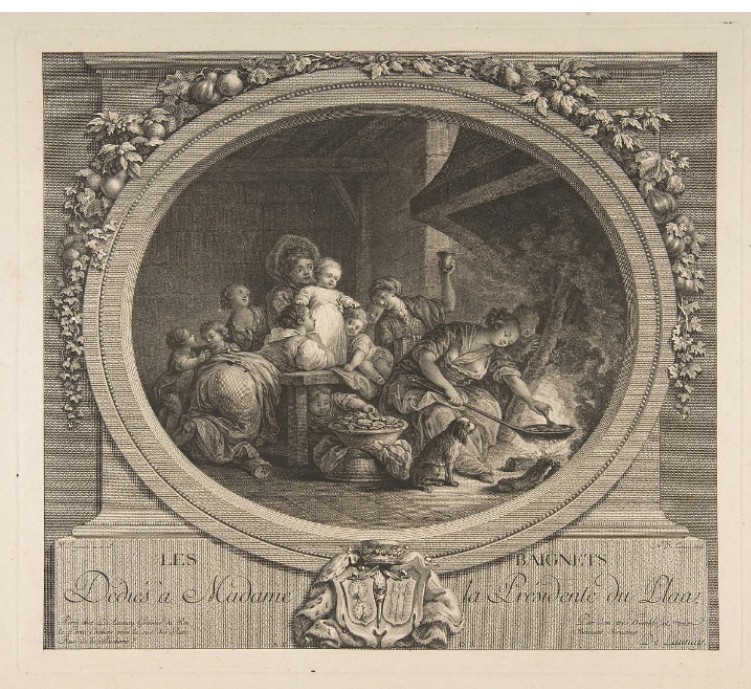

**Figure 1.** Nicolas de Launay (France, 1739–1792) After Jean Honoré Fragonard, *Les Baignets*, 1783, Etching and engraving, 26.7 cm × 30.1 cm, Metropolitan Museum of Art, New-York, Purchase, Roland L. Redmond Gift, Louis V. Bell and Rogers Funds 1972 (1972.539.13).

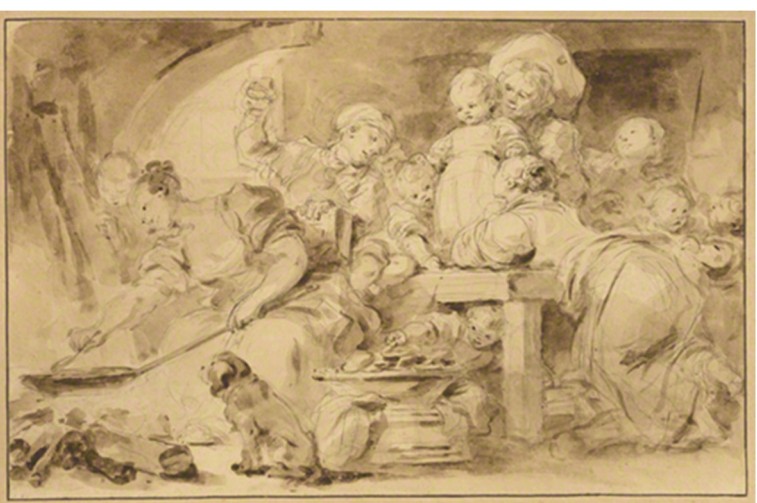

**Figure 2.** Jean-Honoré Fragonard (French, 1732–1806), *Making Fritters (Les Beignets),* approximately 1782, Brush and brown ink over graphite, 24.6 cm × 37.5 cm (9 11/16 × 14 3/4 in.), 2012.4, The J. Paul Getty Museum, Los Angeles.

Using de Launay's print of *Les Baignets* as a case study, and in light of eighteenth-century studies on commerce, economics, botany and originality, I argue that it is precisely the conceptual and physical "marginality" that makes these border decorations important creative spaces for "marginal" artists wishing to communicate a style of their own. I offer a new approach to the study of the role of the engraver/printmaker as active commentators, essential mediators and independent creators.

## 2. The Double Seduction: Framing Decoration and the Print Business

> M. Delaunay, Engraver to the King, of the Royal Academy of Painting and Sculpture, has just published a new print after M. Fragonard, Painter to the King and of the same Academy; its title is *Les Beignets*, and is worthy of the talents of the two Artists; it follows on those which appeared some time ago [ . . . ]; and will complete the six Precious Prints of this genre, which M. Delaunay intends to publish.[5]

De Launay's detailed advertisement announcing the publication of *Les Baignets* relays the importance he attributes to this latest series of prints as well as his hope of increasing its sale as a "pendant", a well-known phenomenon in the print market. Generally, prints made as couples were produced according to clear guidelines: the prints must be the same size, theme, composition, color and "effect".[6] Delaunay adheres to these criteria with *Les Baignets* and its pair, *Dites donc s'il vous plait* (Figure 3), which are of identical dimension, and feature a similar rustic setting, and complementary decorative frame. The frame for *Les Baignets* includes squash, hazelnut, and oak—plants and fruits associated with winter, while the frame of its pendant is laden with summertime crops and ivy (Sheriff 1990, p. 98).

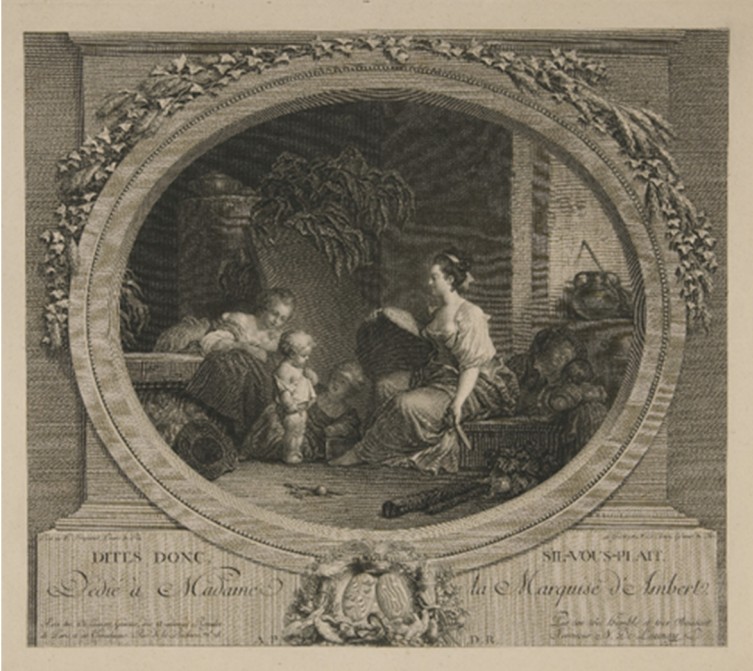

**Figure 3.** Nicolas de Launay after Jean Honoré Fragonard, *Dites Donc, S'il-vous-plait*, 1783, Etching and engraving, 27.5 cm × 31.2 cm, The British Museum, London, © The Trustees of the British Museum.

Engravers acted out of economic interest just like any other business owner. The need to cover the costs of materials and publishing led them to undertake a wide range of sales strategies (Pasa 2020, pp. 23–27). De Launay published dozens of advertisements throughout his life in the *Journal de Paris*, *Journal General de Paris*, *Gazette de France,* and *Mercure de France*, with the aim of arousing consumer excitement and increasing sales (Smentek 2007, pp. 222–24). Special offers on pendants promised him advance payment on future prints, and in turn, buyers of multiple prints were assured of a product that could be displayed as a set (Taylor 1987, p. 516; Rudy 2013, p. 47; Fuhring 2015, pp. 30–35).

The economic motivation behind pendants led Watelet and Levesque to roundly criticize the phenomenon, and specifically the buyers, in *Dictionnaire Des Arts De Peinture, Sculpture et Gravure*:

> [T]he true art lovers look for merit in paintings, and do not hesitate to acquire a precious picture that has no pendant: but those who are only concerned with decoration are not very interested in the merit of the works, & much on their correspondence [ . . . ]. Today, prints are hardly bought except as furniture; an engraver cannot promise himself a sale of a print if he does not accompany it with a corresponding print. As soon as he has engraved a plate, he must hurry to engrave the pendant.[7]

A customer's desire to decorate the walls of his home with art prints, measured in quantity rather than quality, was part of a broader trend in Paris in the second half of the century, when interior design became a widespread and even obsessive pastime (McAllister Johnson 2016, pp. 63–65).

To combat this tendency, the *Dictionnaire*'s authors separated the "true" art lover from the buyer looking for decorative items that were "popular", a dirty word among academicians (Lajer-Burcharth 2018, pp. 23–26; Joullain 1786, pp. 100–2). The fact that prints became cheaper and more accessible made them an important tool in the democratization of art at the same time as they enabled consumers to display status at an attractive price (Bellhouse 1991, pp. 125–27). This might also explain the change from storing prints in albums (where they would be seen by a select few art lovers), and the introduction of the two-dimensional framing device, which created a pleasing visual effect when a pair or more of prints hung together on a wall (and could be seen by many more people) (Auslander 1996, pp. 53–54).

Notwithstanding, the notion of framing was already playing a role in commerce, according to Natacha Coquery: "The decoration and mise-en-scene of the shop became essential components of the marketing strategy" (Coquery 2004, pp. 78–79). While we are all too familiar with the power of visual advertising, its potential as a tool to entice buyers was becoming more common in the eighteenth century. All types of business owners began to arrange their wares more decoratively inside their shops, prepare beautiful shop windows, and invest in outdoor signage (Coquery 2004, pp. 71–72; Walsh 1995, pp. 157–76).

Being inspired by art display and by the new concept of consumer experience, printshop owners habitually accentuated the visual role of the border decoration (Coquery 2004, p. 78). The illusionary frame, besides uniting unrelated fine art prints into pendants or series (Taylor 1987, p. 516), was also used by engravers as a commercial sales strategy in the form of the trade card. First used in the seventeenth century, the trade card gained popularity from 1700 in London and Paris, where it was used to advertise professional artisans and traders specializing in various commodities from food to fabrics, wallpapers, and even, prints (Hubbard 2012, pp. 30–31; A Short History of Trade Cards 1931, pp. 1–6). Although the card's design was usually fixed—a text surrounded by a decorative frame—the cards exhibit charming creativity. For example, Henry Dawkins created an asymmetrical Rococo frame decorated with teakettles and coffee pots for Benjamin Harbeson's copperware business (Figure 4). In another iteration, E. Warner replaced the various kettles and pots with scissors, saws and knives to publicize Henry Patten's razor-making establishment (Figure 5).

In referring to the "double seduction" of the business card, Coquery, alludes not only to the card's text and image as a two-pronged marketing ploy, but also to the fact that the cards themselves quickly became collectibles (Coquery 2004, p. 74). Collected by prominent collectors of prints, the cards were valued according to strict artistic criteria, which led to the blurring between fine art print and advertisement (Hubbard 2012, pp. 40–41), in which both made use of massive decorated frames. Since the frame's mechanical function (to securely hang works) is irrelevant in a two-dimensional trade card, its significance lies in its conceptual function as a demonstration of economic power at a time when carefully carved gilded frames were conspicuous symbols of status (West 1996, pp. 63–78). It is no wonder then that the text inside the elaborate frame should extol not only the product or service for sale but also the seller, maker or professional artisan.

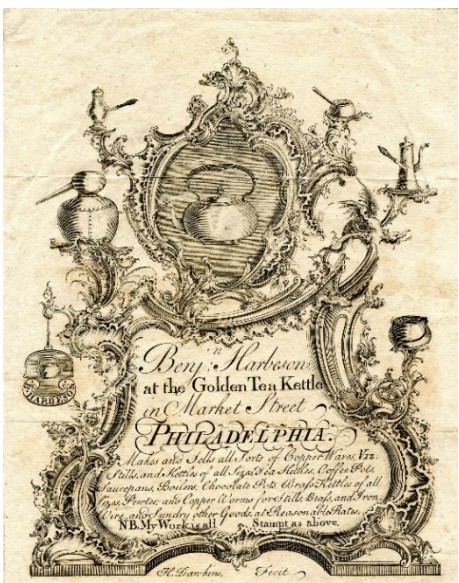

**Figure 4.** Henry Dawkins (England and USA 1753-ca. 1786), *Trade Card for Benjamin Harbeson's copperware business*, 1776, Engraving, 9.65 cm × 7.08 cm, The Winterbur Library, Delaware.

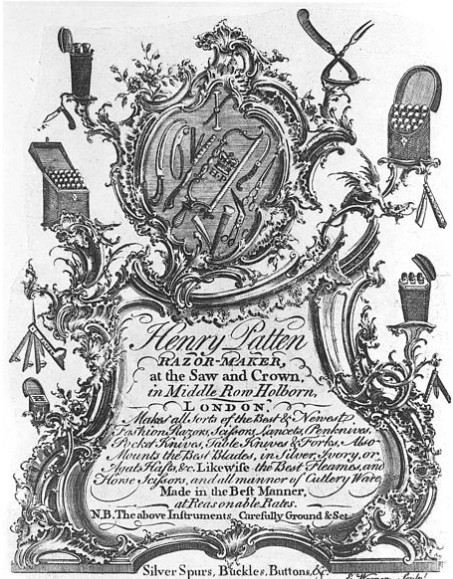

**Figure 5.** Edward Warner (England, active 1750), *Trade Card of Henry Patten, Razor-Maker and Cutler*, 1750 ca., Engraving, ink on paper, 19 cm × 14 cm, Victoria and albert Museum, London.

The conceptual role of the frame was significantly important due to the rising consumer interest in singular items, especially natural materials, which paradoxically led to the manufacture and selling of imitations, which in turn made the terms "quality" and "originality" particularly charged (Berg 2004, pp. 125–32). For some professional artisans and merchants, the frame on their calling cards surrounding a central text that described the superiority and uniqueness of their imitations (Savedoff 1999, p. 354), was an important part of their brand strategy (Berg 2002, pp. 1–30). This was particularly relevant in the print industry, based on the replication of images (which is the medium's particular advantage), which effectively neutralizes the component of originality that is a work of art's defining element (Smentek 2014, p. 148). In the eyes of the Académie's painters, with whom engravers had a complicated relationship in the eighteenth century, the loss of a work's "original" quality, or "aura" to borrow Walter Benjamin's term (see Benjamin 1969,

p. 14), through the reproductive process, led to the perception of the print, and by extension of the engraver/printer, as inferior.[8]

Louis XIV's comparison of the engraver's art to a liberal art that "depends on the imagination of its authors and cannot be subject to laws other than those of their genius",[9] was intended to enhance the standing of the painter-engraver in the market, while at the Académie they remained "Pseudo-Artists" (McTighe 1998, p. 5; Levitine 1984, p. 17). In the eyes of the painter, mass-market prints after a famous painting were a strange hybrid of high art, a popular craft in the service of an imitative practice (Auslander 1996, pp. 53–54; Lajer-Burcharth 2018, p. 23). Thus, the "imagination, originality and genius" lauded by the king in his elevation of the engraver's art, was a vulnerability for the printer.

In an effort to safeguard their interests, the Académie's painters, dependent on the graphic medium for the dissemination of their art, and in an effort to uphold the professional hierarchy among the institutions' members, led them to demand that the roles of all involved in the making of a print be acknowledged on the final product (Mellby 2009); McAllister Johnson 2016, p. 24). Apart from the important matter of copyright (McAllister Johnson 2016, pp. 24–25, 80), these acknowledgements emphasized the marginal and liminal status of the print, and its literal borders became the very place where printmaker/publishers made extraordinary attempts to challenge the existing hierarchy, or at least wrest from the painters some measure of artistic and interpretive freedom.

Recent research on book publishing has drawn attention to the importance of the various stages of a book's production, particularly the technical decisions on artistic matters that influenced and altered the essence of the final product. The cases discussed by (Ann Blair 2016), Peter Stallybrass (2011) and William Slights (1989) among others, testify to an active, even creative process in the addition of para-texts, as Gerard Genette describes in his canonical study of the subject (Genette 1997, pp. 1–2). Because of the historically ambivalent status of the engraver, and despite the similarity between book and print publishing, researchers have continued to view the printmaker as a businessman in the art market,[10] rather than as a resourceful artist and entrepreneur. De Launay was not only a shop owner or a technician. He was a skillful artist, a member of the Denmark Art Academy and he owned a private collection of original paintings and drawings (Taylor 1987, p. 526); all of which enrich our perception of his identity and invite a new perspective of his persona, his career and his art.

Just as the water pump and pipe publicized the plumber's art, the decorative frames on engraver's trade cards promoted their unique brand. For example, William Hogarth's trade card features a frame with an artist sketching on the right and the figure of a classical muse on the left, and a pair of putti above, one of which holds a print. The framing device announces the engraver as a divinely inspired artist in his own right, and challenges the view of the engraver as technician (Figure 6) (Prévost 2013, p. 147). A similar idea is expressed in the frame of Chrêtien de Mechel's card, which includes garlands, putto sketching, a library filled with books and a printing press, which shows the engraver as divinely inspired, learned and possessing technical know-how (Figure 7), defying the division of the "hands" and "mind" common at the time (Diderot 1751, vol. 1, pp. 713–17).

Analogous sentiments can be found also in the fine art printed border decorations of the engravers Sébastien Leclerc (1637–1714), Pierre-Philippe Choffard (1730–1809), Gilles-Marie Oppenordt (1672–1742), Nicolas de Larmessin III (1684–1755) and de Launay. Rather than overt allegories of intellectual prowess or divine inspiration, de Launay's frames demonstrate his astute botanical knowledge, which he cleverly exploits to display his creative talent and unique style, as well as his knowledge of contemporary social issues relevant to his clientele.

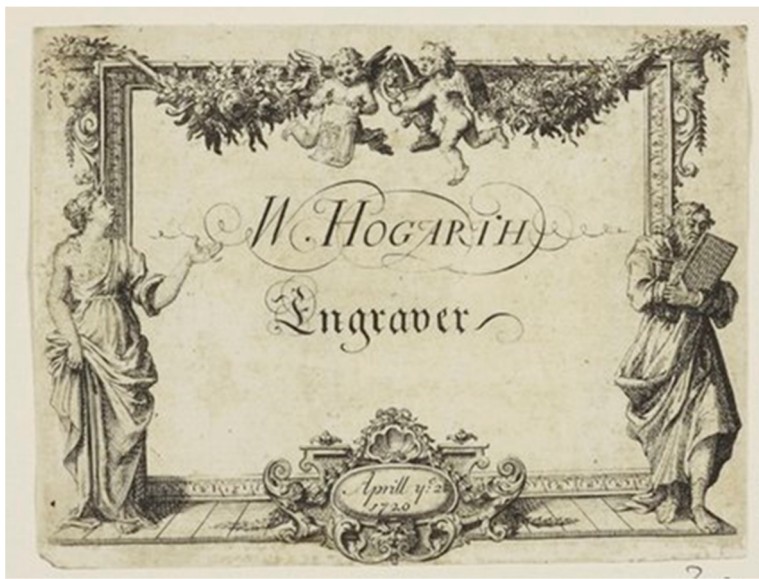

**Figure 6.** William Hogarth (England 1697–1764), *Trade Card*, 1720, Engraving, 7.7 cm × 10.2 cm, Royal Collection London.

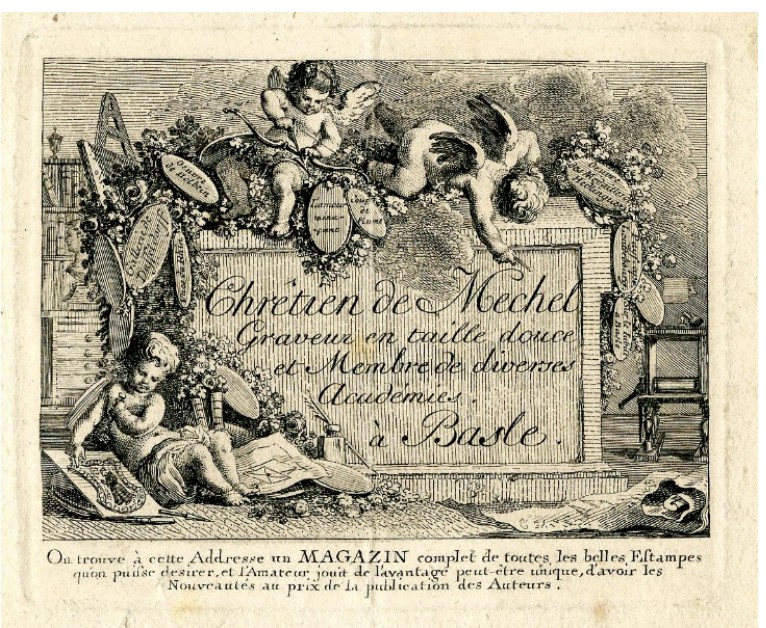

**Figure 7.** Balthasar Anton Dunker (Germany 1746–1807), *Trade card of Chrêtien de Mechel*, 1789, Etching and engraving, 73 cm × 89 cm, The British Museum, London.

### 3. "Every Man under His Vine and Fig Tree": The Clientele, the Real and the Ideal

The happy family frying doughnuts in De Launey's print *Les Baignets* is clearly associated with the winter season (de Goncourt 1865, p. 31), as evidenced by Abraham Bosse's print *L'Hyver*, featuring a family similarly engaged (Figure 8). While both prints show families gathering round a hearth, their surroundings could not be more different. In Bosse's print, an elegantly dressed family fries their Mardi Gras pasties at a hearth decorated with a painting above the mantel, and they will soon sit down to their festive meal at a table covered with a neatly-pressed tablecloth. De Launey's print, on the other hand, depicts a peasant family gathered round a stone hearth in what appears a very humble room. The bare, unadorned walls and floors, and exposed wood-beamed ceiling suggest a sparse country cottage. Yet, these peasants appear neither miserable nor poor. Their loving cuddles, cheery faces and fleshy bodies suggest that they are happy and well

fed, an impression reinforced by the garland of ripe winter fruits that adorns the frame through which we glimpse them.

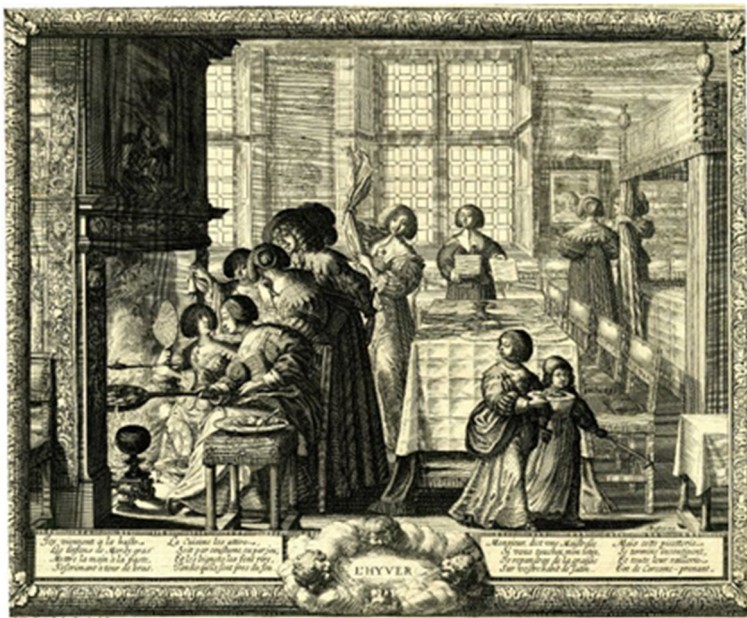

**Figure 8.** Abraham Bosse (French, 1604–1676), *L'Hyver*, 1635–1637, Etching, 26.2 cm × 32.8 cm, The Metropolitan Museum of Art, New-York; Harris Brisbane Dick Fund, 1926.

Each of the frame's components, a festoon of hazelnuts, oak leaves and acorns and ripe squash, relates directly to the peasant way of life. The hazelnut, commonly found in the French countryside, is used to prepare an edible nut-oil (Duhamel Du Monceau 1755, vol. 1, pp. 187–88). The acorn that grows on the common oak (*Quercus Robur*), which can be poisonous for humans, is a popular animal feed, especially for pigs (Duhamel Du Monceau 1755, vol. 2, p. 209). Butternut squash, also known as "winter squash" (Paris 1989, p. 426), is a cheap and simple yet healthy peasant food.[11] De Launey combines these three winter fruits of the land identified with the French peasantry, to form a cozy, protective wreath around the family, as if guarding it from the imagined blustery winds blowing outside. While Fragonard's work on which the print is based, unequivocally relays the idea of the satisfaction of the simple life, De Launay amplifies it through his elegant framing device that alludes to nature's cyclical and organic abundance. To emphasize the point, De Launay imagines the smoke rising from the frying pan in the shape of a tree growing miraculously out of the fire, its branches spreading like a canopy over the mother and her brood as a metaphor of the peasant and the land.

The pendant print, in which a mother demands her son to "say please" when asking for a toy, imparts a similar sentiment.[12] A bunch of juicy beets on the floor in the lower right foreground, and bunches of corn stalks visible in the background, used for insulation or as a bed for animals, hint at the humble foods that more than once saved France's peasantry from famine (Parmentier 1781, pp. 178–80). The stalks of wheat adorning the frame remind of the harvest season and the rewards of agricultural labor (Duhamel Du Monceau 1755, vol. 2, p. 209). This message of nature's bounty as compensation for the bliss of a modest peasant life culminates in the framing device surrounding the print *Conjugal Gaiety*, after a design by the Swiss painter Sigmund Freudenberger. Here, De Launey celebrates nature's autumn abundance with a frame that includes zucchini, carrot, leek, beetroot, celery, leafy greens, and branches of plump quince (Figure 9).

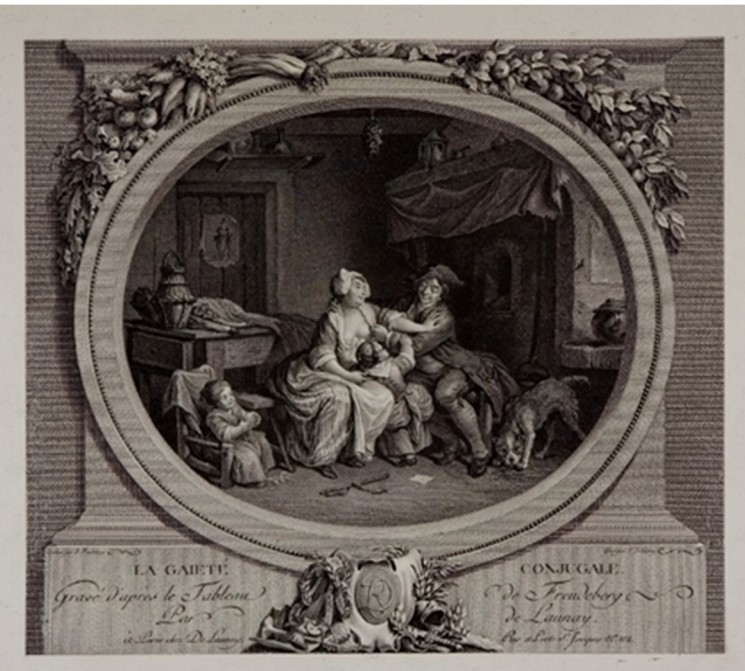

**Figure 9.** Nicolas Delaunay, after Sigmund Freudenberger (Switzerland 1745–1801), *Conjugal Gaiety La gaieté conjugale*, 1783, Etching and engraving, 28.6 cm × 32.5 cm, Harvard Art Museums/Fogg Museum, Gift of Belinda L. Randall from the collection of John Witt Randall, R4793, Photo © President and Fellows of Harvard College.

The remarkable vegetation displays adorning the frames of all six prints in the series[13] suggests a well thought out design program by the engraver. The interiors, as well as the decorations, were unified by the engraver in style and definition, strengthening their relation to one another and emphasizing the simple country life scenery. Yet, while one might construe a framing device that depicts farmers surrounded by the fruits of their labor and benevolent nature as unremarkable and predictable, it is worth considering the social context that drove de Launay to add these alluring framing devices to his prints.

"The farmers. Those who cultivate the lands [ . . . ] produce the wealth and resources essential for the support of the state. That is why the work of the farmer is of immense importance to the kingdom and deserves special appreciation and attention", wrote François Quesnay at the beginning of the entry "Farmers" in Diderot and d'Alembert's *Encyclopédie*.[14] This economic approach underlying Quesnay's physiocratic outlook (de Riquetti et al. 1763; Meek 1960; Vaggi 1987) perceived farming, peasants and farmers as critical to the Empire: "The greater the welfare of the peasants, the greater their ability to produce from their lands and the greater the nation's power is".[15] As France was dependent on its peasants' labor, the images of peasants in these prints were not mirrors of real peasant life, but constituted a mirror of conscience for the whole nation.

According to physiocracy's proponents, France was lagging dangerously behind other European nations in agricultural development, a situation that did not bode well for the Old Regime. Realizing the national consequences of agricultural neglect, the government initiated a panicked response to encourage farming (Heller 2009, p. 30). The palace recruited academics to implement practical education for gardening and agriculture (Gillispie 2004, pp. 335, 357–70; Sexauer 1976, p. 503), and various researchers, with the government's backing (Spary 2014, p. 33), began to write instruction manuals of various kinds, from planting fruit trees to recipes for rice porridge, corn pastries, and potato bread, as for example Henri-Louis Duhamel du Monceau's books on the care and use of local trees (Gillispie 2004, pp. 360–66; Duhamel Du Monceau 1755). Naturally, the print shops in Paris produced, published and disseminated these manuals (Stearn 1962, p. 148).

An enormous gap separated the idyllic plan of building an autarkic system and increasing local agricultural production and its actual implementation, which was complicated by long intervals between peak crop-producing seasons, as happened approximately 1740, and periods of drought, like the one that began approximately 1770, which may have precipitated the French Revolution (Farr 2008, pp. 41–44). The eighteenth century in France thus became synonymous with persistent famine and disappointing agriculture, and forced its peasantry to resort to buying food at markets and supplementing their meager incomes through loans (Emsley 2014, pp. 65–66; Heller 2009, pp. 28–29).

The seemingly newfound awareness of the country's dependence on the peasant class manifested itself in a rise in popularity of images of peasant life in the art market (Farr 2008, p. 41; Schnapper 1990, pp. 33–34). Evidence of the aristocracy's love of scenes of rustic life can be found in the dedication of the print of *Les Baignets* to Madame la President du Plaa, who owned the original work by Fragonard.[16] The prevalence of peasant scenes in the print market suggests a "broad" audience. However, their actual buyers were mostly wealthy Parisians whose knowledge about real peasant life and experience was infinitesimal (Smentek 2007, p. 225; McAllister Johnson 2016, p. 41).

"Graceful and elegant peasants led a graceful and elegant life. It is not that the style contradicts the scene, but the scene described contradicts reality" (Sheriff 1990, p. 102), writes Mary Sheriff of the dissonance between the image and reality (See Castriota 1995). In this light, it is no surprise that de Launay accentuates the sense of harmony in his village-life prints. His decorative floral frames surrounding scenes of farm life symbolize the well-being, stability, and satiety of his customers more than of the peasants they frame. There is an irony as well, considering the lands many of these peasants farmed belonged to the wealthy print owners who hung the prints in the Paris homes (Ruff 2015, p. 48; Seaton 1982, p. 63).

In his research on the pastoral influences in art and literature in the eighteenth century, Alan Ruff remarks on the political significance of representations of pleasant flowering vegetation, which he perceives as part of the deliberate attempt to stimulate patriotic sentiment and convey economic success (Ruff 2015, p. 77). Clearly, the regime supported the positive connotations associated with the rustic life disseminated in the prints, and as the revolution progressed, the practice intensified (Simpson 2005, p. 275; Emsley 2014, pp. 65–66), as can be seen in the increasingly fecund borders in de Launay's prints from the 1780s. Furthermore, the increasing abundance on the frames reflects another widely accepted practice at the time: the display of vegetables, fruits, and cereals from local produce in the public sphere (Spary 2012, pp. 259, 278; Du Monceau 1768, pp. 203–4). The foods were not for sale, nor were they distributed to the hungry. The government relied on the power of visual abundance to relay a positive message, even if, in reality, this was not the case.

A similar idea is conveyed in the decorations surrounding allegorical royal portraits, such as the marital alliance between Louis XVI and Marie Antoinette (Figure 10). A variety of ripe fruits surrounds the oval portrait in addition to the traditional olive branches. Generally a symbol of fertility (and the hope of future heirs through the marital alliance), in this instance, the accompanying text, which mentions abundance, evokes propaganda motifs related to economic stability (See Castriota 1995). A double portrait of Louis XVI and his benevolent ancestor Henri IV features a pair of cornucopia overflowing with grains, grapevines, squash, and acorn—the very same produce that de Launay replicated in his framing devices of happy peasant life (Figure 11). This clearly propagandist portrait seeks to attribute to Louis XVI the very qualities that earned Henry IV the title "Le Grand" (Buisseret 1989, pp. 1–3, namely, his great skill in handling periods of famine and his concern for the poor peasants) (Péréfixe de Beaumont 1661, pp. 154–55.)

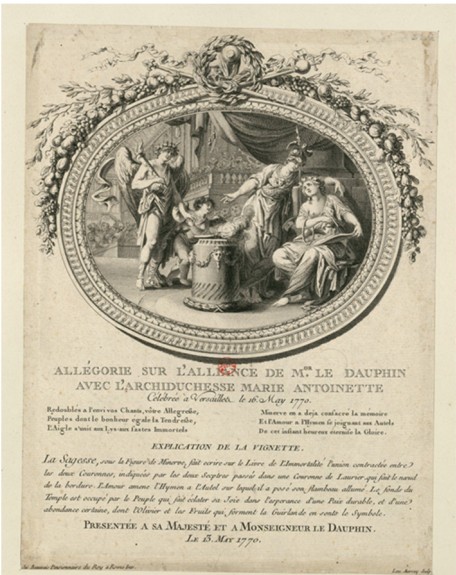

**Figure 10.** Pierre-Laurent Auvray (France, 1736–1781), after Jacques-Philippe de Beauvais (France, 1739–1781), *Allégorie sur l'Alliance de Mgr le Dauphin avec l'Archiduchesse Marie-Antoinette, Célébrée à Versailles*, 16 May 1770, Engraving, 16 cm × 19 cm, Bibliothèque Nationale de France, Département Estampes et Photographie; Collection De Vinck (histoire de France, 1770–1871).

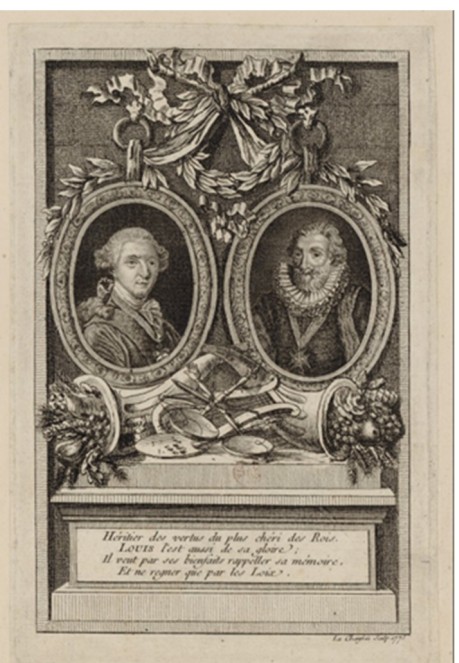

**Figure 11.** La Chaussee (France, unknown dates), *Portraits de Louis XVI et de Henri IV dans deux Médaillons Suspendus au-dessus de Deux Cornes d'Abondance*, 1775, Engraving, Bibliothèque nationale de France, Département Estampes et Photographie; Collection Michel Hennin, Estampes Relatives à l'Histoire de France. Tome 109, 9476–9572.

De Launay further embellishes the theme of the regime's care and nourishing of its people by accentuating the bosom of the mother frying donuts at the hearth, a change he made to Fragonard's sketch during the engraving process. This alteration evokes the notion of breastfeeding and the idea of nurturing land, glorified in the eighteenth century by philosophers, moralists, and physicians alike,[17] that was part of a rich world of imagery

related to perfecting the changes in the family institution, as Carol Duncan and Ewa Lajer-Burcharth have shown.[18]

Images of a mother breastfeeding or with exposed breasts mirrored not only medical or moral attitudes toward motherhood, marriage and child rearing, but served the ideas promoted by the government (Bellhouse 1991; Ventura 2018). The entry "Woman" in Diderot and D'Alembert's *Encyclopédie* from 1756 reveals the political-economic context that may have underpinned the image of the mother:

> "occupied with the governing of her family, she rules over her husband through kindness, over her children through sweetness [ . . . ] her house is the abode of religious sentiment, filial piety, [ . . . ] order, interior peace, gentle sleep and of health: thrifty and settled, she thereby avoids passions and needs; the poor man who comes to her door is never turned away."[19]

A mother's governance over matters of religion, order, peace, and especially welfare extends to the notion of the role of the nurturing country that was rooted in Western culture and was commonly illustrated in the form of a bare-breasted woman holding a cornucopia (Joyce 2014).

While these themes have been addressed in the research dealing with French eighteenth-century paintings, their treatment by the engravers and publishers has been ignored in the scholarship. These graphic images contain critical information about the reception of current ideas, and the roles of the engravers in perfecting and disseminating them. De Launay's grouping of images into a unified series and his emphasis on the nurturing and care of a stable society is an alluring strategy through which he speaks to his aristocratic clients in their language, at the same time as he lays the ground for propagating his own.

## 4. The Nature of the "True Artist"

De Launay created his agriculture-themed frames specifically for a clientele who saw their own comfort reflected in the happy scenes of peasant life surrounded by nature's bounty. The interest in visual representations of food can be traced to a preoccupation with healthy cooking and nutrition in Europe, especially France, in the eighteenth century (Spary 2012, pp. 243–89). Medical knowledge had begun to recommend healthier, more nutritious and natural foods such as vegetables, fruits, milk, and eggs, which explains the appearance of these products in the contemporary visual culture (Spary 2014, pp. 125–28, 158–59; Mennell 1996, pp. 34–35).

The preference for "natural" over "processed" foods, and attention to qualities such as "fresh" and "pure" (Spary 2014, p. 95), were attempts to rebrand the "artificial" life of the city.[20] Thus, among the aristocracy, food was linked with social status and capitalist consumer culture alongside concerns about health.[21] In the salons, there were discussions about food and cooking together with discussions on art, philosophy, and literature. An interest in both imported and regional foods was considered to be indicative of a person's status and taste. One could say that in the eighteenth century, the scientific, aesthetic and sentimental interest in nature influenced the contemporary food culture and expressed a longing for the unique and the authentic in general.

Hence, images of maternal breasts garlanded with foods fresh from the field reflected the flourishing scientific research that sought to connect man and nature. The Swedish botanist and scientist Carl Linnaeus had a decisive role in this discourse (Schiebinger 1993; Johnson 2011, pp. 171–92), as evidenced by the burgeoning field of botany that reached it apex in the second half of the eighteenth century (Williams 2001; Woudstra 2000). De Launay's print series thus expresses not only the relationship between man and nature in terms of the agricultural economy and the Motherland's nourishment of her people; it unequivocally embodies the contemporary enthusiasm for botany, which was hugely popular also in art circles (Hyde 2005, pp. xvii, 59, 77).

In the frontispiece of the *Encyclopédie*, the figure of "Botany" is seated next to the sciences of "Optics", "Chemistry" and "Agriculture" (May 1973, p. 164), with "Music",

"Painting", "Sculpture" and "Architecture" in a group slightly above to the left (Figure 12). The allegorical figures representing Botany and Painting gaze in opposite directions, visualizing the opposing fields of the Art and the Sciences. However, the two groups are linked visually by the gentle touch of Painting's toe next to the seated Botany, hinting at the connection between science and art.

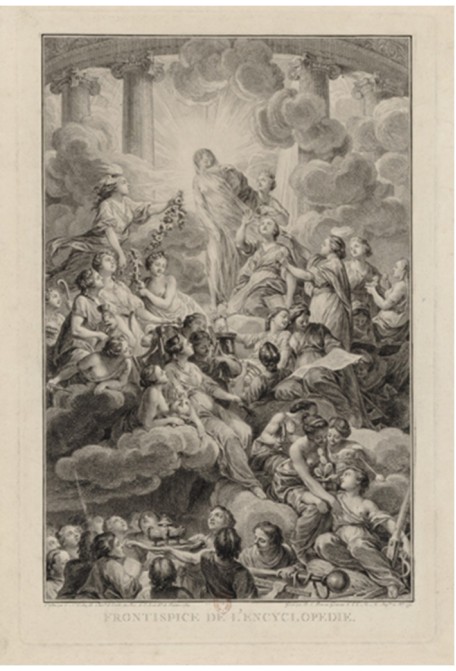

**Figure 12.** Benoît Louis Prévost (France, 1735?–1804?), after Charles-Nicolas Cochin fils (France, 1715–1790), *Frontispice de l'Encyclopédie ou Dictionnaire Raisonné des Sciences, des Arts et des Métiers*, 1772 (originaly 1765), Engraving, 33.7 cm × 21.9 cm, Bibliothèque Nationale de France, Paris; Recueil Collection Michel Hennin.

In previous centuries, botanical research focused on plants' medicinal benefits. In the eighteenth century, it investigated their biological and visual structure. Engravers collaborated with scholars to create botanical prints,[22] with many engravers eventually becoming botanical experts in their own right.[23] Nicolas Francois Regnault, a draughtsman and engraver who also created engravings after paintings by Fragonard, published a botanical book with his wife, Geneviève de Nangis-Regnault, which included some three hundred illustrated plates in addition to copious text (Portalis and Béraldi 1880–1882, vol. 3, p. 386).

De Launay's frames exhibit a similar scientific expertise. His attention to detail, precise copying of every leaf, fruit, and stalk, are consistent with what Nicolas Green calls the "true artist" of the time, who must, "have an encyclopedic first-hand knowledge of all the components making up the natural world. [ . . . ] it was essential to master [ . . . ] close-ups of rock shapes and the texture of foliage and bark. [ . . . ] A late product of the obsession in late eighteenth-century scientific thinking" (Green 1990, p. 113).

In addition to his meticulous rendering of the flora, de Launay shows his deep understanding of plants' specific features and uses. A good example are two later pendant prints de Launay's published on Christmas 1790 as part of his ongoing series (*Gazette de France* 1790, p. 516). The first, *Education Fait Tout* (Figure 13), shows a mother watching her older children play with, or educate (as the title implies), a pair of cocker spaniels dressed in costume (Simons 2015; Milam 2007, 2015). The decorative frame surrounding the scene is adorned with horse chestnut branches (*Aesculus Hippocastanum*), which only appeared in forests and Paris boulevards from the seventeenth century (Duhamel Du Monceau 1755), and were not common in art and certainly not decorative art.

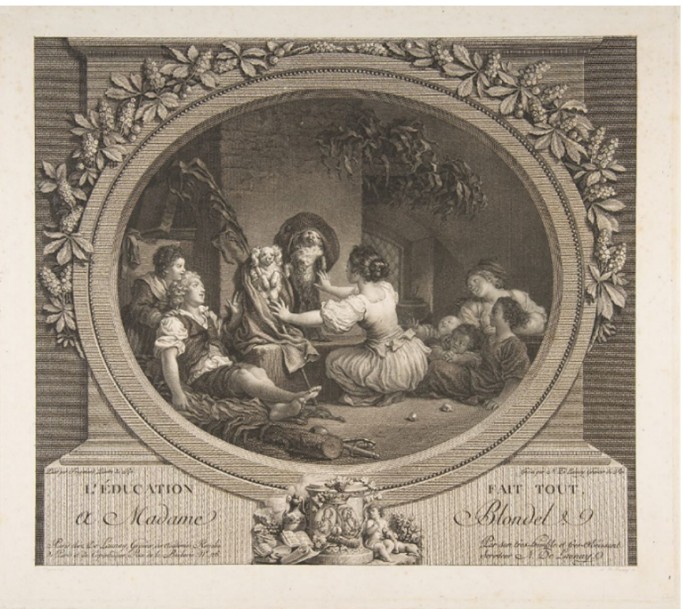

**Figure 13.** Nicolas de Launay after Jean Honoré Fragonard, *L'Education Fait Tout,* 1790, Etching and Engraving, 26.9 cm × 30.6 cm, Metropolitan Museum of Art, New-York, Purchase, Roland L. Redmond Gift, Louis V. Bell and Rogers Funds 1972; 1972.539.16.

Depictions of the tree's beautiful branches appear in Duhamel Du Monceau's *Traité des Arbres et Arbustes* (Figure 14). Du Monceau, and Michel Adanson in his *Familles des Plantes* (Adanson 1763, pp. 380–81; Duhamel Du Monceau 1755, vol. 1, p. 296; Du Monceau 1780, pp. 41–42), separately note the tree's "beautiful pyramids of white flowers",[24] and the fact that peasants would use the tree's chestnuts for kindling fires in winter and as feed for farm animals (but not for the horses) (Duhamel Du Monceau 1755; Du Monceau 1780, pp. 41–42; Adanson 1763, pp. 380–81). De Launay's choice of this flowering branch for the frame surrounding the scene of rustic enjoyment shows his expert botanical draughtsmanship and familiarity with the tree's use by the peasantry.

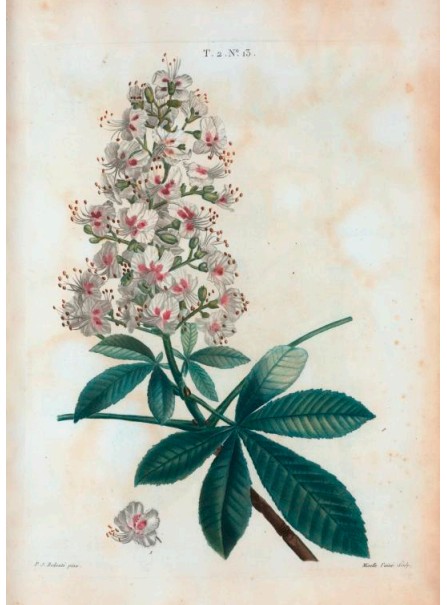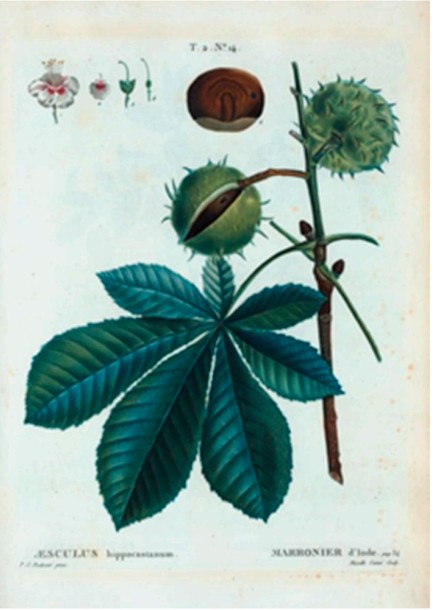

**Figure 14.** Duhamel du Monceau (France 1700–1782), Redouté, Pierre Joseph, (France 1759–1840), *Traité des arbres et arbustes que l'on cultive en France en pleine terre* (Paris: Duhamel du Monceau 1801), Rare Book Division, The New York Public Library.

The frame surrounding the pendant piece titled *Le Petit Predicateur* (Figure 15), features an unconventional tree called the chaste tree (*Vitex agnus-castus*), which is characterized by oval flowers and was known for its healing properties (Rozier and Claret de La Tourrette 1787, vol. 3, pp. 537–39; Miller 1786–1789, vol. 8, p. 60; Duhamel Du Monceau 1755, vol. 2, pp. 357–58). Already from the fifteenth century, herbalists held that an extract made from this tree encouraged the production of breast milk and helped suppress excessive sexual desire (Hobbs 1991, p. 22; Schellenberg 2001, pp. 134–37). This hypothesis was given scientific credence by M. Geoffroy in his medical treatise of 1743, in which he wrote that chaste tree is "very useful in suppressing the fires of lust [ . . . ], and dispels the dirty images that come during sleep".[25]

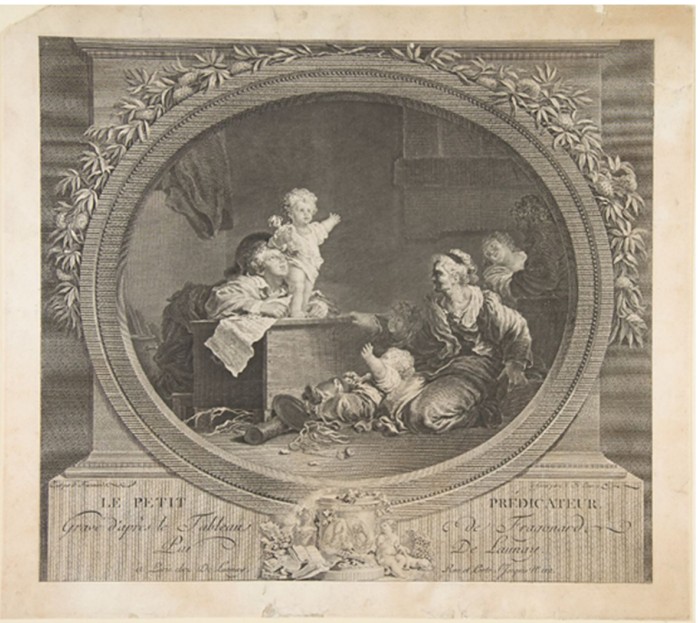

**Figure 15.** Nicolas de Launay after Jean Honoré Fragonard, *Le Petit Predicateur,* 1790, Ething and engraving, 26.8 cm × 30.2 cm, Metropolitan Museum of Art, New-York, The Elisha Whittelsey Collection, The Elisha Whittelsey Fund, 1958.

It is possible that De Launay chose this specific tree branch for the frame due to this unique feature, as his commentary on a narrative detail in the main scene. At the center of the print is a little boy who delivers a sermon while standing atop a wooden trunk. His father kneels behind him and holds on to his feet in case he loses his balance.[26] At the right, the little boy's mother looks in the direction of her son, and next to her is a looming male figure in the shadow who appears to grab her (Rosenberg 1987, pp. 465–66). The chaste tree in the frame demonstrates de Launay's own thinking on the subliminal message of the scene at the center.

Beyond the affinities between frame and image, de Launay's beautiful renderings of squash, hazelnuts and oak, horse chestnuts, chaste tree, quinces, ivy, lilac and vines form a veritable collection of botanical prints. Delaunay ultimately offers his customers two collections for the price of one that reflect his great technical expertise and intellectual and scientific knowledge as well as his customers' desire for elegantly framed copies of popular works by well-known artists and for fashionable botanical prints (Stearn 1962, pp. 138–40; Hall 1986). His series allows his clientele to enjoy the rustic life and beautiful nature without actually having to care for either (Spary 2000, p. 17). These prints show de Launay to be a true creative artist who uses the margins of his "canvas" to create a new type of artwork. Acknowledging this creative platform raises questions about the historically deferential position of the printmaker in relation to the painter and the competitive relationship between the original painting and the reproductive print.



### 5. To Touch with One's Eyes: Visual Illusion and Sensory Experience

The style of de Launay botanical frames is very different from the decorative art at the time, such as by Pierre Ranson who was one of the most prolific French decorative designers in the second half of the century. Ranson also includes butternut squash, oak, quinces, wheat, garden leaves, carrots in his designs, but their overall effect is artificial (Figure 16). The combinations do not show a seasonal rationale and their scale is unfaithful to reality. The overall design is reminiscent of heraldic emblems rather than actual nature.

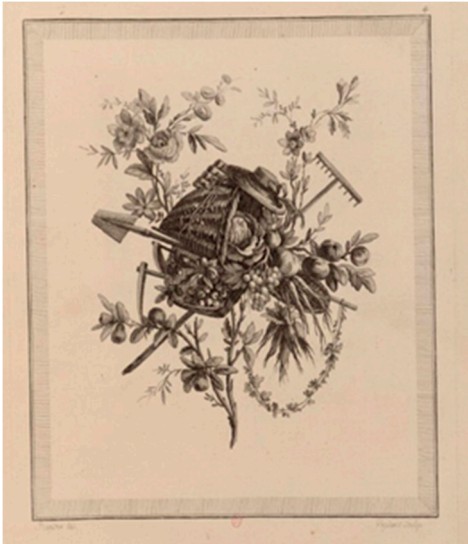 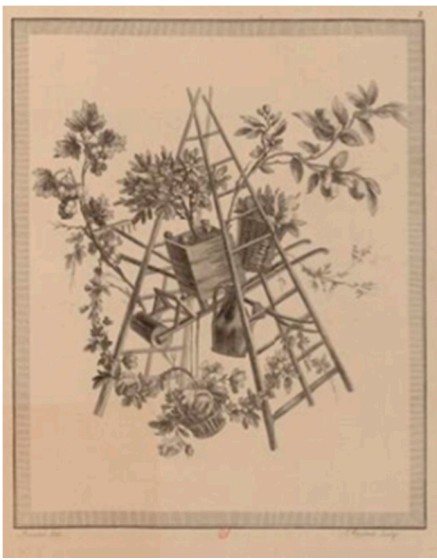

**Figure 16.** Pierre Ranson (1736–1786), *Oeuvres Contenant un Recueil de Trophées, Attributs, Cartouches, Vases, Fleurs Ornemens Et Plusieurs Desseins Agréables pour Broder des Fauteuils; Composés et Dessinés par Ranson, et gravés par Berthaut et Voysard* (Paris: (Ranson 1778)).

De Launay's designs compared to Ranson are like a blossoming tree versus a carefully arranged flower bouquet. This conflict between "natural" and "artificial" did not only play out in the graphic medium in the eighteenth century. It was inherent in all attempts at nature's portrayal (Elias 1994, pp. 3–4). For example, in the second half of the eighteenth century, the "natural" garden concealed the landscape architect's role in its creation.[27] The tension was also at the heart of the aesthetic "Picturesque" that sought to capture man's encounter with nature, even though "nature" was a manmade conceit (Wiebenson 1978). A similar duality is found in the "Fabrique"—the fabricated buildings (in gardens or paintings) that resembled ancient ruins overtaken by nature (Watelet and Levesque 1756, vol. 6, pp. 351–52; Symes 2014, p. 120). This tension between the imitation and the real pervaded the marketplace as noted above in the discussion about frames on trade cards advertising imitation goods and reproductive prints with visual hints to their originality and authenticity. Ranson's decorations, featuring realistic looking three-dimensional plants, flowers and fruits, were also a site of conflict, one that de Launay wished to resolve.

De Launay's frames resemble classic decorations and correspond with the Doric pedestal at the base answering the "Goût grec" aesthetics of his clients and oppose the grotesque fabricated picturesque (Mitchell and Roberts 1996, pp. 44, 48, 65). Furthermore, the *tromp l'oeil* of natural plants create a distinction between the painting at the center and the oval frame with seemingly real branches adorning it. An example of this deceptive mingling of the artificial and the real can be seen de Launay's print *La Consolation de l'Absence* (Figure 17). De Launay, no longer faithful to the original (Figure 18), heightens the emotional reading by replacing the mirror with a painting of Cupid. More germane to the present study is the elaborate garland surrounding the painting, which leaves the viewer to wonder whether it is a woodcarving or made from actual flowers, and importance of imitation and its spectacle.

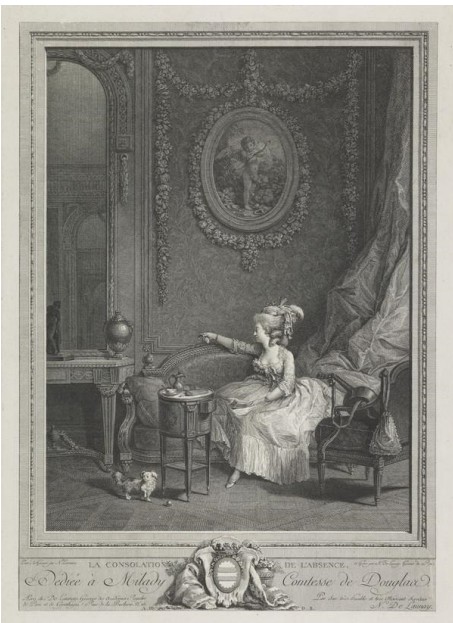

**Figure 17.** Nicolas Delaunay, after Nicolas Lavreince (Sweden, 1737–1807), *La Consolation de l'Absence*, 1785 ca.,Etching and engraving, 35 cm × 24.7 cm, National Gallery of Art, Washington D.C.

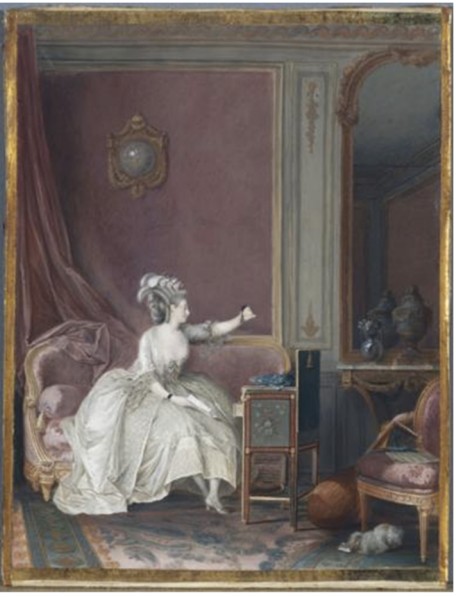

**Figure 18.** Nicolas Lavreince (Swedish, 1737–1807), *La Consolation de l'absence*, 1785, Gouache, Vélin, 25 cm × 20.5 cm, Musée Cognacq-Jay, le goût du XVIIIe, Paris.

This special emphasis placed on deceiving the eye in the liminal space was a well-known practice in late medieval and Renaissance Books of Hours (Figure 19) (Kaufmann and Kaufmann 1991, p. 47), where the decoration marked the boundary between the sacred center and its mundane surroundings (Carr 2006). The very real appearance of the ornamental plants and flowers (Kaufmann and Kaufmann 1991, p. 49), succulent squash, insects, snails and butterflies reflect the manuscript's function: the book's owner would hold the volume while praying, his/her fingers touching the margins that look as though they are alive, enhancing the religious experience as Kaufman and Kaufman argue (Kaufmann and Kaufmann 1991, p. 53). The medievalist Michael Camille argues that the ornate frame decorations were intentionally sensory; however, they were answering the

book owner's material need to attain valuable and original objects with his or her own hands, as a sign of status.[28]

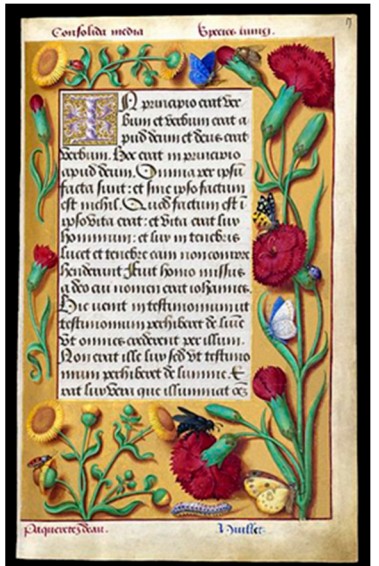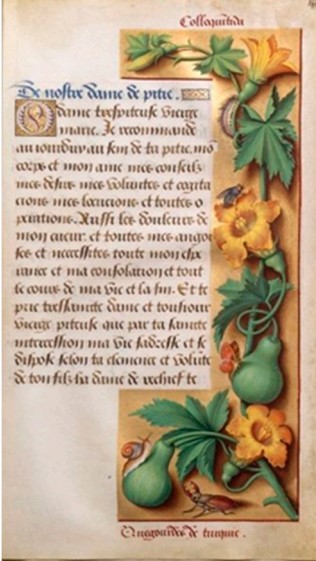

**Figure 19.** Jean Bourdichon (France, 1457–1521). *Les Grandes Heures d'Anne de Bretagne*, 1503–1508, Manuscript, h: 33 cm, Bibliothèque Nationale de France; Département des Manuscrits.

Rousseau conveys similar sentiments about the powerful effects of touching and handling precious ephemeral objects, with reference to his own collection of prints: "From time to time, I look at my [print] album next to the fireplace. It distracts me from my torments and comforts my sufferings".[29] Even if eventually hung on a wall, the print beckons close looking, which can be done only by holding it in one's hand and touching its edges. The senses of sight and touch evoked through *tromp l'oeil* in the print emphasized the role of senses in the Enlightenment movement.[30]

Indeed, the Enlightenment relies on a human's use of the five senses to explore reality and understand it directly (Scarry 2005, p. 97). In her book, *The Painter's Touch*, Ewa Lajer-Burcharth argues that Chardin, Boucher, and Fragonard, three paradigmatic French painters of the eighteenth century, gave priority to the sensory experience in their work in response to their audience's need for materiality (Lajer-Burcharth 2018, pp. 3–7). A similar approach emerges in Emma Spary's book on food, which argues that sensory experiences informed the French food culture and that the exploration of taste underlies the cooking revolution that took place during the Enlightenment.[31]

Coquery, too, claims that the senses were critical in the consumer world, as shop owners encouraged customers to hold a product in their hands, touch it, smell it (Coquery 2004, p. 79). Similarly, their business cards, designed to be held, evoked the memory of touching, smelling, hearing, or tasting a coveted product (Hubbard 2012, p. 33). I contend that this sensory infrastructure is part of de Launay's style, which he honed for his clientele seeking sensory experiences, as well as to emphasize his own creation as vital as opposed to the inanimate center.

However, the growing appetite for the sensory experience in the eighteenth century was also the engraver's weakness. Diderot compares the engraver to a translator: "Engravers are in fact writers, wishing to translate a poet's language to another one [ … ] when the engraver is intelligent, one look at the print will be enough to *sense* the original painter's style".[32] However, he finds the final product wanting: "We must admit, that in comparison to the painting, the role of the engraving is quite cold".[33] De Launay and his fellow engravers were mediators bound to fail, translators of "untranslatable" content, to use Ricœur's terminology (Ricoeur 2006, pp. 30–39). That is, their prints could not portray the sensory dimension, scale or color of a painting; and even the best burin engravings

could not achieve the brush strokes or texture of paint applied to a canvas. Hence, engravers were ultimately perceived as subservient to the painter, intermediaries who could not give viewers an "authentic experience" (McAllister Johnson 2016, pp. 23–24).

De Launay could not, nor did he intend to create a perfect copy of the original painting. Indeed, compared to the life-like margins, the image at the center appears lifeless and static. To borrow Rene Magritte's famous words, "ceci n'est pas une Peinture"; it is a mere imitation, a representation of a painting.[34] However, de Launay's illusionistic framing devices convey the power of imitation. De Launay admits that a print after a painting is not a painting, but his own original contribution to the final product challenges the creative hierarchy by drawing attention to the mediated experience and singularity of the reproductive work. Along the same lines, Richard Cullen Rath explains that paradoxically, in the eighteenth century, the importance of sensory experience was expressed precisely through its mediation, which was frequently more accessible than direct experience (Rath 2019, p. 206). Unfortunately, Rath excludes the fine art print from his discussion, a communication medium whereby the engravers were agents of mediation between the inaccessible painting and the audience.

That De Launay copied paintings from his private collection, such as *Le Petit Predicateur* and *Le Education Fait Tout*, indicates not only his genuine love for art, but also the physical and sensory intimacy with the original that motivated him to replicate it.[35] Likewise, his botanical imagery exemplifies his view of himself as a mediator of authenticity, as Lisa Gasbarrone points out with regard to Jean-Jacques Rousseau's writings on botany (Rousseau 1780–1782; Rousseau 1823). Rousseau was fundamentally preoccupied with "nature" because for him, "the natural object itself must always be seen as the 'original,' more striking and authentic than any imitation of it could be" (Gasbarrone 1986, p. 7). For the people of the period, Rousseau among them, nature represented "naturalness", in the sense of the authentic essence of an entity, as D'Alembert writes in the entry "Nature" in his *Encyclopédie* (d'Alembert 1765, vol. 11. p. 40).

Rousseau believed that texts about botany, including his own, hindered direct affinity with the source, a sentiment I believe is equivalent to the case of the print. Nevertheless, he chose writing to resolve the dilemma. He wrote a variety of botanical texts with the aim of "bridging the gap between words and things".[36] For Rousseau, words at once demarcated and mediated between man and nature, similar to how a print simultaneously distances and facilitates the viewer's encounter with the original painting. In this light, De Launay's frames of botanical representations can be viewed as a natural, original and beautiful bridge owning up to his role as necessary mediator.[37]

## 6. Inside Out

Upon the completion of the painting *Israelites Gathering the Mana* in 1639, Nicolas Poussin wrote to his patron: "Once You are in receipt of your painting, if you like the idea, I ask you to adorn it with a frame; this is necessary so that when viewing it in all its parts, the rays of the eye are focused and do not become distracted".[38] Aware of external motives and competition, Poussin further notes that he prefers a very humble golden frame, one that will not interfere with the center, that will simply disappear.[39] Even though Poussin's request is aesthetic, the subtext clearly recounts the instrumental role of frames in the hands of different agents. While painters need the frame to protect their product, and while patrons influence the art they purchase by adorning it with specific frames; the use of framing decorations in order to establish a unique signature style that correlates with the artist's client is a remarkable strategy. As Nicoals de Launay's hybridist artworks demonstrate, in the hierarchal print market of the eighteenth century, this method was specifically clever.

Literally placing the paintings on a pedestal in his prints, de Launay honors them by enveloping them with illusionary picture frames that distinguish their status. Having done so, he is free to use the margins as important alternative *centers* for his own creativity, as described by Victor Turner in his writings on liminal spaces (Turner 2008, pp. 95–96);

(Otto 2016, p. 140). The frame becomes a strategic calling card alluring his clientele with subtextual ideas and in vogue aesthetics. While maintaining his professional integrity, de Launay asks that his works be seen as "creations in their own right" (Gombrich 1984, p. 251), to use Ernst Gombrich words on marginal décor, hinting that their marginality is no less fascinating than the center. Although they might appear as threatening the prestigious painting at the center, they are neither a "Trojan horse" nor a "parasite." On the contrary, as this article demonstrates, these frames provide the possibility for dialogue between the center and its margins, between artists and viewer.[40]

**Funding:** This research received no external funding.

**Conflicts of Interest:** The authors declare no conflict of interest.

## Notes

1    The misspelling of the word *Beignets* is in the title of the original print.

2    In the lower right corner, "Par son très humble and très obeissant servituer." McAllister Johnson (2016).

3    My deep gratitude goes to Dr. Yuval Sapir, Curator of Herbarium at the Steinhardt Museum of Natural History, The George S. Wise Faculty of Life Sciences, Tel Aviv University, for his knowledgeable insights and help with categorizing the vegetation throughout this research, and to Alexandra Dvorkin for her contribution and insights on the botanical illustrations.

4    In the upper right corner of the rectangular plinth on which the image rests: "H Fragonard inv et del". these letterings indicate the medium of the original was drawing, "Peint" was added when copying a painted image.

5    "M. de Launay, Graveur du Roi, de l'Académie Royale de Peinture & de Sculpture, vient de publier une nouvelle estampe d'après M. Fragonard, Peinture du Roi & de la même Académie; elle a pour titre Les Beignets, elle est digne des talents des deux Artistes; elle fait suite à celles qui ont paru il y a quelque temps sous le [ . . . ]; elle sera suivi cette année de trois autres de la même grandeur & du même format qui compléteront les six Estampes précieuses de ce genre, que M. de Launay se propose de publier". *Mure ercdu France* (1783, pp. 137–138). The print was the third in a series of six created by de Launay, who decided to pair every two prints in the series as pendants, as his advertisements indicate.

6    Watelet and Levesque write "conformite dans [ . . . ] l'effet", which I translate as having a similar effect on the viewer. Watelet and Levesque (1792, vol. 5, pp. 1–2); McAllister Johnson (2016, pp. 1–2, 58); Taylor (1987, p. 516).

7    "[L]es véritables amateurs de l'art ne recherchent dans les tableux que leur merite, & ne negligent pas d'acquerir un tableux precieux que n'a pas de pendant: mais ceux qui ne s'occupent que de la decoration, sont peu difficiles sur le merite des ouvrages, & beaucuop sur leur correspnondence. [ . . . ]. aujourdui qu'on n'achete guere des estampes qu'en qualite de meubles, un graveur ne peut se promettre un debit sur d'une estampe, s'il ne l'accompagne pas d'un estampe correspondante. des qu'il a grave une plache, ul faut qu'il se hate graver le pendant." Watelet and Levesque (1792, vol. 5, pp. 2–3).

8    Printmakers did not have equal rights in the Académie and were perceived in principle as subordinate. Carlson (1984, pp. 25–26); McAllister Johnson (2016, pp. 78–80).

9    "[D]épend de l'imagination de ses auteurs et ne peut être assujetti à d'autres lois que celles de leur génie, [ . . . ] en doit être entièrement libre" (Société de l'histoire de l'art français 1862, p. 262).

10    To mention just a few, Mathis (2015); Smentek (2007).

11    Spary (2014, pp. 33, 95); Sheriff (1990, p. 101). In general, squashes and the rest of the gourd family stood out in period recipes, as they offered prolonged satiety at a lower cost.

12    The original painting is lost. There are similar versions such as "The Class Teacher". *Mure ercdu France* (1783, pp. 137–38). Portalis (1889, pp. 187, 299); Wildenstein (1960, p. 302; cat. 468, 469); Goubert (1968, p. 600).

13    Nicolas de Launay after Jean Honoré Fragonard, *L'Heureuse Fecondite*, Etching and engraving, 26.9 cm × 30.5 cm, Metropolitan Museum of Art, New-York; Nicolas de Launay after Jean Baptiste Le Prince, *Le bonheur du Ménage*, 1778, engraving and etching, 29 cm × 32.4 cm, The British Museum, London; Nicolas Delaunay and Jean-Louis Delignon after Sigmund Freudenberger, *La Félicité Villageoise*, ca. 1770–1780, etching and engraving, 29.3 cm × 34 cm, Davison Art Center, Wesleyan University.

14    "Fermiers, sont ceux qui afferment & font valoir les biens des campagnes, & qui procurent. Les richesses & les ressources les plus essentielles pour le soûtien de l'état; ainsi l'emploi du fermier est un objet très-important dans le royaume, & mérite une grande attention". Quesnay (1756, vol. 6, pp. 528–29).

15    "Plus les laboureurs sont riches, plus ils augmentent par leurs facultés le produit des terres, & la puissance de la nation." Quesnay (1756, vol. 6, p. 533).

16    More prints from the series include the names and coat of arms of aristocrats such as Louis Gabriel, Marquis de Véri Raionard, Madame Marquise d'Ambert.

17    Corsini and Matthews-Grieco (1991, p. 49); see Ventura (2018, pp. 3–80).

18  The importance of "motherly feelings" and enjoyment in child-rearing had a didactic function to reflect and shape the social perception of parenting. For more on this, see Duncan (1973); Lajer-Burcharth (2007).

19  "Renfermée dans les devoirs de femme & de mère, elle consacre ses jours à la pratique des vertus obscures: occupée du gouvernement de sa famille, elle règne sur son mari par la complaisance, sur ses enfants par la douceur, sur ses domestiques par la bonté: sa maison est la demeure des sentiments religieux, de la piété filiale, de l'amour conjugal, de la tendresse maternelle, de l'ordre, de la paix [ . . . ], l'indigent qui se présente à sa porte, n'en est jamais repoussé [ . . . ]." Corsembleu de Desmahis (1756, vol. 6, p. 475).

20  Spary (2014, pp. 89–90, 93, 114–57, 244–45). On eighteenth-century food produce and market, see Jones (1993).

21  Martin (2011, p. 143). Further reading on cooking, food and status in the French Enlightenment, see Bickham (2008, pp. 73–78). Mennell (1996, pp. 69–82, 108–26).

22  Ruff (2015, p. 70); Stearn (1962, pp. 138–44). The Enlightenment paid special attention to nature and to man's direct sensual and emotional relation with it. Teute (2000, p. 319); Hyde (2005, pp. 122–26).

23  See for example, Duhamel Du Du Monceau (1801).

24  "Belles Pyramids de Fleurs Blanches". Duhamel Du Monceau (1755, vol. 1, p. 295).

25  "Les uns disent qu'il est très-utile pour réprimer les feux de la Luxure [ . . . ], & dissipe les sales imaginations qui viennent pendant le sommeil." Geoffroy (1743, vol. 5, section 2, p. 75).

26  *Aubert sales catalogue*, no. 74. Quoted in Rosenberg (1987, p. 466).

27  Ruff (2015, pp. 77–79). See also Hunt (1992, pp. 171–85); Hays (2017).

28  Camille (1992, pp. 153–57). See also Orth (1996).

29  "Je parcours de tems mon portefeuille au coin de mon feu; cela me distrait de mes maux et me console de mes misères. Je sens que je redeviens tout à fait enfant." Rousseau (1823, vol. 29, sct. 1, p. 140).

30  See Purnell (2017), and Addison's publications: Addison (1712, no. 411–21).

31  Spary (2012). See also Korsmeyer (1999, pp. 38–67).

32  "Le graveur en taille-douce est proprement un prosateur qui se propose de rendre un poète d'une langue une autre [ . . . ]. En qualité de le style de traditeur d'un peinture, le graveur doit montrer le talent et de style de son original [ . . . ]. Lorsque le graveur a été un homme intelligent, au premier aspect de l'estampe, la manière du peintre est sentie". Diderot ([1765] 1984, p. 314).

33  "Il faut avouer aussi qu ' à côté de la peinture, le rôle de la gravure est bien froid." Diderot ([1767] 1876, vol. 11, p. 367).

34  *La Trahison des images*, 1928–1929, oil on canvas, 60.33 cm × 81.12 cm, Los Angeles County Museum of Art. Stoltzfus (2013); Prang (2014, pp. 420–21).

35  Taylor (1987, p. 526). Artists, like botanist, sought to create from unmediated experience, seeking to see, touch, and smell the original, copying from reality and often visited galleries to do so. Green (1990, p. 113).

36  "Rousseau responds to the challenge of botany ingeniously and energetically [ . . . ], He produces a variety of botanical texts, each one an attempt to fill the sign, or at least to *bridge* the gap between words and things." Gasbarrone (1986, p. 8).

37  Further emphasis on how botanic floral imagery related to the artificial representation of nature, see Kalba (2012).

38  "Je vous supplie, si vous le trouvez bon, de l'orner (le tableau) d'un peu de corniche, car il en besoin, afin que, en le considérant en toutes ses parties, le rayons de l'œil soient retenus et non point épars au dehors, en recevant les espèces des autres objet . . . ". Marin (1982, p. 18).

39  To use Derrida's definition of the "Parergon", Derrida (1979, p. 21).

40  This article is based on my PhD dissertation on Framing Decoration in the long eighteenth century's reproductive art. My deepest thanks go to Prof. Gal Ventura, who was not only a wonderful dissertation adviser, but is a true mentor; to Dr. Sharon Assaf for her great contribution and her thoughts; and to Prof. Daniel M. Unger for constructing this brilliant Special Issue.

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
