# Peer review of "Cutting Edges: Professional Hierarchy vs. Creative Identity in Nicolas de Launay’s Fine Art Prints"

_arts_

Round 1
Reviewer 1 Report
The author of the article under review has an excellent insight to share with the readers. While frames and framing is a usual subject of serious theoretical scrutiny, the objectives of frame designers is a fresh turn of this topic. The study of De Launay’s case shows that while adding illusionistic frames to their prints reproducing somebody other’s work, the engravers provided to themselves the space for self-expression and self-assertion. Moreover, the author demonstrates that this was not just the matter of identity and professional pride, but of economic strategy. The most revealing is that the engraver took liberty to change the original composition by the famous painter (Fragonard), to comment on it by his inventive additions, to compete with him and thus to assert his place in the arts as that of equal to painters. This thesis is very well argued, I was especially impressed by the analysis of botanical knowledge of the artist and of his informed use of it in his work. As a whole, the essay gives a glimpse into a scarcely explored territory.
I highly recommend the essay for publication, and I am going to make a few comments and suggestions that can help to make it more smooth for the reader.
- The relation between the De Launay’s print and Fragonard’s original composition is to be clarified better. If the author is sure, that the original was the drawing and not a missing painting after this drawing, this is to be explicitly stated and Fragonard’s practice to produce independent drawings and to provide them for the use of engravers referenced. The very comparison between the drawing and the print has not been pursued to the end in the article: the author has not mentioned the stylistic changes, in particular the introduction of the space of the house, which is excluded from the drawing and is very well defined in the print. Moreover, this domestic setting is similar throughout the series of prints including compositions by different authors. The description of the situation of the print as the third in the series of six (in the original of the advertisement) on the one hand and being a part of a pendant pair with another print, on the other, is also not clear from the text.
- Close to the end of the article, the author mentions De Launay’s private collection of drawings – this fact could be used effectively in discussion of his business.
- Some editing needed when the author describes the concept of Motherland: Roma is not precisely Motherland, moreover, the Roman and French terms for Motherland is patria or patrie. So that it is better to replace the political connotations (Motherland) by natural ones (nurturing land, country and nature).
- The author arranges a comparison of De Launay’s frame with Ranson’s decorative compositions, opposing them as “natural” type of composition similar to picturesque garden, to an “artificial” arrangement. The problem with this paragraph is that in the 18th- century terminology, just the contrary, Ranson’s compositions are in “pittoresque” (asymmetrical) genre. I would recommend to explain De Launay’s vegetal garlands as more neoclassical ones in spirit than Ranson’s “grotesques”, and this will be in consent with the architectural design of De Launay’s frames reminiscent of Doric pedestal (if possible, it would be nice to add an explanation of this motif as well) .
- The spelling and the translation of the French original quotations (where the original is provided) is not always absolutely accurate (see lines 41-44, 69-73 and the related fn).
- Referencing: the author is evidently perfectionist. Evident observations like that a fashionable interior decoration was a sign of status, need not be referenced to scholarly contributions, unless these contributions state something special to mention. The objective and meaning of extensive bibliographical note 34 remained obscure to me. The note 45 is nice, and necessary, but it is more natural for the reader to meet it either in the beginning of the article or in the beginning or in the end of the section.
In conclusion: this insightful essay is to be published and I hope that employing these comments will contribute to its quality.
Author Response
Dear Reviewer,
My deepest thanks for your insights and contributions towards making this paper better, and for highlighting specific changes to be made.
I am enclosing a revised version of the text. You will see that I embedded most of the comments.
you can also see specific answers here, bellow.
Many thanks and all the best!
- The relation between the De Launay’s print and Fragonard’s original composition is to be clarified better. If the author is sure, that the original was the drawing and not a missing painting after this drawing, this is to be explicitly stated and Fragonard’s practice to produce independent drawings and to provide them for the use of engravers referenced. The very comparison between the drawing and the print has not been pursued to the end in the article: the author has not mentioned the stylistic changes, in particular the introduction of the space of the house, which is excluded from the drawing and is very well defined in the print. Moreover, this domestic setting is similar throughout the series of prints including compositions by different authors. The description of the situation of the print as the third in the series of six (in the original of the advertisement) on the one hand and being a part of a pendant pair with another print, on the other, is also not clear from the text. Response: You are right, and I added some information in the introduction to clarify it better.
2. Close to the end of the article, the author mentions De Launay’s private collection of drawings – this fact could be used effectively in discussion of his business. Response: I agree, and did mention it. Unfortunately, we don’t know enough about the collection itself (overall quantity and quality), and de Launay's motivations in collecting for reproduction? for future sales? Or for the love of art? I hope to deepen my understanding of this collection and its purposes in the near future.
3. Some editing needed when the author describes the concept of Motherland: Roma is not precisely Motherland, moreover, the Roman and French terms for Motherland is patria or patrie. So that it is better to replace the political connotations (Motherland) by natural ones (nurturing land, country and nature). Response: I have made this change, as the semantics are crucial. The concept of the mothering nature and the idea of motherland was very strong at the time, and played a role in the separation between Woman=Nature and Man=Culture – Unfortunately I had to cut this out of the final version of this paper.
4. The author arranges a comparison of De Launay’s frame with Ranson’s decorative compositions, opposing them as “natural” type of composition similar to picturesque garden, to an “artificial” arrangement. The problem with this paragraph is that in the 18th- century terminology, just the contrary, Ranson’s compositions are in “pittoresque” (asymmetrical) genre. I would recommend to explain De Launay’s vegetal garlands as more neoclassical ones in spirit than Ranson’s “grotesques”, and this will be in consent with the architectural design of De Launay’s frames reminiscent of Doric pedestal (if possible, it would be nice to add an explanation of this motif as well) . Response: I think the sentence was not clear. I agree with this remark and changes the wording. I also agree that there is more to be said on the classic style. However, I am currently writing a paper on the sculptural and classic elements in these prints, with an emphasis on Fragonard's original pieces and the relation with de Launay (which will also answer better to point no. 1 of the review.). Therefore, it is excluded from this text.
5. The spelling and the translation of the French original quotations (where the original is provided) is not always absolutely accurate (see lines 41-44, 69-73 and the related fn). Response: Thank you, I corrected the quotation.
6. Referencing: the author is evidently perfectionist. Evident observations like that a fashionable interior decoration was a sign of status, need not be referenced to scholarly contributions, unless these contributions state something special to mention. The objective and meaning of extensive bibliographical note 34 remained obscure to me. The note 45 is nice, and necessary, but it is more natural for the reader to meet it either in the beginning of the article or in the beginning or in the end of the section. Response: You will see some of these changes in the text.

Reviewer 2 Report
This is a clearly written essay with interesting contributions to the interpretation of framing elements as creative expressions of ideas. It is thorough in its use of secondary sources and the engagement with a vast range of visual sources supporting the arguments of the author. At times, I thought the argumentation would have been more productive if tightly focused around the Nicolas de Launay prints and the interplay with Fragonard’s paintings — the meanings of both within each medium and how they altered when framed in the prints. However, that might be explored in a subsequent article. As research that seeks to reinstate the independent artistry of a print maker with a substantial reputation and body of works who is little known in his own right, this essay merits publication. Print making is increasingly an area of interest for scholars and remains under researched. It is exciting to see interpretive engagement with elements of art that have remained peripheral to our understanding of historical ways of seeing and representing the visible world in the eighteenth century.
Author Response
Dear Reviewer,
My deepest thanks for your insights towards making this paper better, and for your kind words.
I completely agree there is more to be said on the specific relation between Fragonard and de Launay, and I'm currently working on a paper that will concentrate on Fragonard's original pieces copied by de Launay, highlighting questions of translation and appropriation, with regards to style and medium. Therefore, these issues were excluded from this text.
Many thanks and all the best!